# Ubiquitination-Proteasome System (UPS) and Autophagy Two Main Protein Degradation Machineries in Response to Cell Stress

**DOI:** 10.3390/cells11050851

**Published:** 2022-03-01

**Authors:** Yanan Li, Shujing Li, Huijian Wu

**Affiliations:** Key Laboratory of Protein Modification and Disease, School of Bioengineering, Dalian University of Technology, Dalian 116024, China; yananeli@mail.dlut.edu.cn

**Keywords:** cell stress, ubiquitin, ubiquitin–proteasome system, autophagy, endoplasmic reticulum stress, unfolded protein response

## Abstract

In response to environmental stimuli, cells make a series of adaptive changes to combat the injury, repair the damage, and increase the tolerance to the stress. However, once the damage is too serious to repair, the cells will undergo apoptosis to protect the overall cells through suicidal behavior. Upon external stimulation, some intracellular proteins turn into unfolded or misfolded protein, exposing their hydrophobic regions to form protein aggregation, which may ultimately produce serious damage to the cells. Ubiquitin plays an important role in the degradation of these unnatural proteins by tagging with ubiquitin chains in the ubiquitin–proteasome or autophagy system. If the two processes fail to eliminate the abnormal protein aggregates, the cells will move to apoptosis and death. Dysregulation of ubiquitin–proteasome system (UPS) and autophagy may result in the development of numerous diseases. This review focuses on the molecular mechanisms of UPS and autophagy in clearance of intracellular protein aggregates, and the relationship between dysregulation of ubiquitin network and diseases.

## 1. Introduction

Under environmental stimuli, cell maintains the cellular homeostasis by increasing tolerance to damage and repairing damaged macromolecules and organelles. However, when the cell damage exceeds the capacity of adaptive response, the cell initiates the apoptosis and death [1]. Common stimuli include heat stress, oxidative stress, hypoxic stress, and DNA damage, which will quickly destroy the protein-folding ability of the endoplasmic reticulum (ER) and induce accumulation of misfolded and unfolded proteins in the ER [2]. To deal with this situation, the stressed cells activate the unfolded protein response (UPR) [3,4,5]. UPR then restores the endoplasmic reticulum homeostasis and protein refolding pathway in three main ways. Firstly, protein synthesis is reduced to mitigate further aggregation of unfolded proteins. Secondly, the protein-folding ability in ER is enhanced by inducing chaperone molecules and folding enzymes’ expression. Thirdly, abnormal proteins are cleared by the endoplasmic-reticulum-associated degradation pathway (ERAD), which transports the unfolded and misfolded proteins out of the ER by retro-translocation and is further degraded by the ubiquitin–proteasome system and autophagy [3,4,6]. If neither of the above processes can alleviate ER stress caused by the accumulation of abnormal proteins, the apoptosis or death of cells will be triggered [1]. There are three UPR pathways in mammal cell, the activation of which is determined by three trans-membrane proteins located in ER membrane, namely activating transcription factor 6 (ATF6), PKR-like ER kinase (PERK), and inositol requiring enzyme 1 (IRE1) [5]. In a physiological cell context, ATF6, PERK, and IRE1 bind to glucose-related proteins (GRP78) and are inactivated. When folded proteins gather in the endoplasmic reticulum, GRP78 binds to the unfolded protein, and ATF6, PERK, and IRE1 are released and activated; then the unfolded protein signal is transduced across the endoplasmic reticulum to the cytoplasm and nucleus (Figure 1). The PERK/ATF4 axis induces the expression of chaperones and proteins involved in autophagy [7], apoptosis, and redox homeostasis. The IRE1/X-box-binding protein 1 (XBP1) axis promotes the transcription of UPR gene sets which are related to the full folding and secretion of proteins. Activated ATF6 induces the expression of chaperone proteins, such as XBP1 and ERAD-related proteins.

Ubiquitin is a classical molecular label for protein degradation. When the misfolded proteins and unfolded proteins are tagged with ubiquitin, these proteins will be degraded in proteasome or lysosome [8,9]. Ubiquitin not only plays a role in UPR, but also participates in the activation of cellular stress responses. UPS and autophagy are two independent system in protein quality control (PQC). However, there is more evidence about the connections between UPS and autophagy [10]. In this review, we first briefly summarize the ubiquitin pool, UPS, and autophagy and then discuss, in detail, various examples of coordination and crosstalk between them and dysregulation of UPS and ubiquitin-mediated autophagy pathways in human diseases, neurodegenerative diseases, and cancer, in particular.

## 2. Ubiquitin Pool

Ubiquitin is a highly conserved small protein molecule containing 76 amino acids and is founded in all eukaryotic cells. Furthermore, ubiquitin molecules are very stable and will not be changed when exposed to acid stress or heat shock. These two properties give ubiquitin molecules an important role in the cell’s defense against the abnormal accumulation of proteins caused by stress. According to the lysine residues in the ubiquitin molecule, the ubiquitin chain can be divided into seven isoforms, namely K6, K11, K27, K29, K33, K48, and K63 [5]. Ubiquitin moieties can be conjugated through N-terminal methionine residue (M1) [11]. In addition to polyubiquitination, there are monoubituitination and multi-monoubituitination. Polyubiquitination can also be divided into homotypic polyubiquitination and heterotypic polyubiquitunqition. Different ubiquitin-chain links function as different regulators; among them, the K48 chain is the main mediator of degradation.

UPS and autophagy are two pathways related to protein degradation, and the activation of the two processes depends on the ubiquitin pool in cells. Intracellular ubiquitin consists of monomer ubiquitin and substrate-bound ubiquitin. Most deubiquitinases (DUBs) play an important role in maintaining the ubiquitin homeostasis in cells by releasing monomer ubiquitin from ubiquitinated protein, which is degraded by proteasomes or lysosomes and recovers the ubiquitin level [12]. DUBs, meanwhile, maintains protein stability, edits ubiquitin chains, processes ubiquitin precursors, and removes non degradative ubiquitin signal. However, the stable state of intracellular ubiquitin still needs appropriate synthesis to compensate for the basal ubiquitin reversal. There are four ubiquitin genes: *UbC*, *UbB*, *UbA52*, and *UbA80*. *UbC* and *UbB* encode polyubiquitin, while *UbA52* and *UbA80* encode ubiquitin and two small ribosome fusion proteins [5]. Therefore, small ubiquitin molecules are all derived from the primary translation products of ubiquitin genes, which would be further hydrolyzed by ubiquitin-specific protease. Almost all of the polyubiquitin genes are stress regulatory genes with heat-shock elements in their promoters [13]. Similarly, the expression of *UbC* and *UbB* is upregulated under various stresses [14,15]. NF-E2-related factor 1 (NRF1) [16], NRF2 [17], SP1 [15], and Heat-Shock Transcription Factor 1 (HSF1) [18] were identified as transcription factor of *UbC*, and these transcription factors may upregulate *UbC* transcription under cell stress. Several studies have shown that mice with targeted *UbC* gene knockout would die at 12.5–14.5 days during embryonic life period, accompanied by severe liver defects. Embryonic fibroblasts from embryonic mice with *UbC* gene knockout showed slower growth, premature senescence, increased apoptosis, delayed cell-cycle progression, and decreased levels of ubiquitin homeostasis. Thus, homeostasis of the ubiquitin pool is important for cell survival, especially under severe stresses [19].

## 3. UPS

### 3.1. Mechanism of UPS

Unfolded and misfolded proteins are toxic to cells and must be eliminated quickly and efficiently. To achieve this, mechanisms controlling the protein quality have been formed to clear the denatured proteins in eukaryotic cells. Eukaryotic cells have developed specific protein quality-control mechanisms to recognize and process these unnatural proteins. One important defense mechanism is the specific elimination of these proteins by the UPS [12]. Ubiquitin-26s Proteasome system is an ATP-dependent non-lysosomal protein degradation mechanism in cells. In this process, proteins can be degraded efficiently and selectively under cell stress, such as cyclin, p21, p53, c-Jun, c-fos, and other structural proteins. The degradation of proteins via ubiquitin-26s proteasome system is a multistage process in presence of ubiquitin, three ubiquitin enzymes, and proteasomes. There are three kinds of ubiquitin enzymes, including E1 (ubiquitin activating enzyme), E2 (ubiquitin conjugating enzyme), and E3 (ubiquitin ligase). Ubiquitin E3 ligase determines the specificity of the substrate, which can be divided into HECT, ring-finger, and U-box ubiquitin E3 ligase according to the biochemical characteristics and structural differences. Chaperones and proteases play a key role of PQC when protein aggregates occur. Molecular chaperones promote the folding of new peptides and refolding of misfolded proteins to inhibit the aggregation of protein through recognition of hydrophobic patches of misfolded protein and unfolded protein under cellular stress [20]. However, when the unnatural proteins can no longer be refolded, it is necessary to identify these misfolded proteins and degrade them through the ubiquitin–proteasome pathway. Misfolded and unfolded proteins are firstly recognized by chaperones, such as Hsp70 and Hsp90, which are mainly chaperones in PQC, and both of them utilize co-chaperones to recognize and bind substrate [21]. Then unnatural proteins either refold in an ATP-dependent manner or are labeled by ubiquitin under the action of chaperon-bound ubiquitin E3 enzymes, such as STIP1 Homology and U-Box Containing Protein 1 (CHIP), BAG Cochaperone 1 (BAG1), and Scythe (Figure 1).

### 3.2. Molecular Chaperones and Ubiquitin E3 Ligases Involved in Cell

Molecular chaperones and ubiquitin E3 ligases are key factors in the degradation of misfolded proteins via the proteasome pathway. CHIP is a typical representative and the most thoroughly studied ubiquitin E3 enzyme related to protein quality control. CHIP has the function of both molecular chaperone and ubiquitin E3 ligase. Studies have shown that CHIP cooperates with BAG-1 to transform the activity of Hsc/Hsp70 molecular chaperone system, from promoting protein folding to promoting protein–ubiquitin-mediated degradation. Interestingly, CHIP also mediates the ubiquitination degradation of Hsp70 (Figure 1). Misfolded cystic fibrosis transmembrane conductance regulator (CFTR) [18,19,20], glucocorticoid hormone receptor (GR) [21], Erb-B2 Receptor Tyrosine Kinase 2 (ErbB2) [22], and Pael receptor (Pael-R) [22] have been identified as substrates for CHIP. Misfolded Integrin, Pdr5, and HMG-CoA reductase (HMGCR) are the substrates of ERAD-related E3 Ligase Der3, gp78, and SCF Fbx2 [23]. To date, nearly 40 ERAD-related ubiquitin E3 ligases have been identified in mammals. The major ubiquitin E3 ligases are summarized in Table 1.

## 4. Autophagy

### 4.1. Mechanisms of Autophagy

Besides UPS, autophagy can also protect the cell by removing toxic protein aggregates and damaged organelles from cytotoxicity pressure. Autophagy was originally thought to non-selectively degrade long-lived proteins and organelles for nutrient cycling and energy production [9]. Later studies have shown that protein aggregates can be removed by selective autophagy. The process of autophagy is usually divided into four stages: induction, nucleation, elongation and substrate isolation, and fusion with lysosomes (Figure 1). The initial signal of autophagy usually comes from a variety of stress conditions, such as hunger, hypoxia, oxidative stress, protein aggregation, and oxidative stress. The target of these signals is ULK1 complex (composed of ULK1, ATG13, ATG101, and FIP200) [55,56,57]. Moreover, mTORC1 is a key regulator of autophagy [58]. As a receptor of energy and nutrition, the activity of mTORC1 is inhibited after starvation [55], which leads to dephosphorylation of ATG13 and activation of ULK1 complex [59]. Subsequently, ULK1 complex phosphorylates autophagy protein BECN1 and promotes it to form a class PI3K complex (including PIK3R4, BECN1, ATG14, and VPS34), which mediates the formation of autophagy vesicles. The extension of autophagy depends on two ubiquitin-like connection systems. In the first one, ATG16L1 binds with ATG5–ATG12 to form the active ligase-like complex ATG12–ATG5–ATG16L1. In the second conjugation process, LC3-pro is hydrolyzed by ATG4B and expose the C-terminal glycine and then became LC3-I. LC3-I combined with PE to form LC3-PE (LC3-II) under the catalysis of E1-like ATG7, E2-like ATG3 and E3-like ATG16L1–ATG5–ATG12 complex [60]. LC3-II anchors to the autophagy vesicle and promotes the extension of autophagy vesicle by recruiting other membrane structure, and then wraps the substrate in the cytoplasm to form autophagosome. In the mature stage of autophagy, autophagosomes fuse with lysosomes under the action of GTPase Rab7 and Synaptosomal-Associated Protein (Synaptosomal-Associated Protein, SNAP) receptor Syntaxin 17 (Syntaxin 17, STX17) to form autophagy lysosomes (Autolysosomes). Finally, protein aggregates and damaged organelles are degraded by lysosome proteases and sent to the cytoplasm for cell reuse [56] (Figure 1). The activity of ULK1 is also regulated by 5′-AMP-activated protein kinase (AMPK) [59,61]. AMPK phosphorylates ULK1 and phosphorylated ULK1 phosphorylates Raptor, resulting in a decrease in the binding of Raptor to mTORC1 and a decrease of mTORC1 activity. In addition, AMPK can also phosphorylate Raptor directly [61] (Figure 1).

### 4.2. Cargo Receptors of Autophagy

The cargo receptors p62, NBR1, Optineurin (OPTN), Toll Interacting Protein (TOLLIP) and TAXBP1 act as bridges between the ubiquitin chain and LC3, helping with the formation of aggregates [55,62,63] (Figure 1). Moreover, p62 binds with ubiquitin or poly-ubiquitin chain through its ubiquitin-associated domain (UBA) and then transfers the substrate to autophagosome and lysosome through LC3-interacting region (LIR) domain and PB1 domain. Although p62 could bind to both K48 and K63 chains, it has a higher affinity for K63 chains. The intracellular level of p62 depends on transcriptional regulation and post-translational autophagy degradation. The transcription of p62 is regulated by NRF2, ras/MAPK pathway, JNK/c-Jun pathway and some compounds [55]. NBR1 and p62 form a complex to promote the formation of aggregates. Overall, p62 is the major driver of ubiquitin condensate formation. NBR1 recruited TAXBP1 into ubiquitin aggregates formed by p62 [62]. Although all three receptors interact with FIP200, TAXBP1 is the main driving force for recruitment of FIP200 and degradation of p62–ubiquitin aggregates through autophagy [63]. TNF receptor associated factor 6 (TRAF6) catalyzes K63 ubiquitination of mTOR and activates mTORC1 through its interaction with p62 in nutrient-activated cells [64,65].

### 4.3. UPR and Autophagy

During endoplasmic reticulum stress, UPR regulates autophagy in different ways [4]. On the one hand, the UPR signal can regulate autophagy by regulating the activity of AKT, mTORC1, and AMPK (Figure 1). For example, activated PERK induces the translation of ATF4, and it further increases the expression of sestrin 2 (SESN2) and DNA-damage-induced transcript 4 (DDIT4). SESN2 and DDIT4 directly inhibit the activity of mTORC1 [37,57] (Figure 1). At the same time, ATF4 can inhibit the activity of AKT and then inhibit the activity of mTORC1 through the DDIT3/TRIB3 pathway [66]. Meanwhile, AMPK can also be activated by PERK signal pathway; activated AMPK cannot only directly activate ULK1, but it also inhibits the activity of mTORC1. The IRE1 signal pathway activates RPS6KA3, and RPS6KA3 also activates ULK1 through activating AMPK [57] (Figure 1). JNK is one of the key regulator of IRE1 signal axis, and BECN1 is the main downstream regulatory factor of JNK. After JNK is activated, Bcl-2 becomes phosphorylated, and that destroys the interaction between BECN1 and Bcl-2 and induces the autophagy of tumor cells [67,68] (Figure 1). ATF6 can suppress autophagy by activating AMPK [69]. UPR cannot only regulate protein activity of autophagy related proteins, but it can also regulate the transcription of key proteins in autophagy. For example, hypoxia-induced endoplasmic reticulum stress can induce ATF4 and CHOP-mediated upregulation of LC3 and ATG5 [70]. Furthermore, eIF2a-kinases, PERK, ATF4, and DDIT3 activate the transcription of a series of autophagy-related genes, including LC3, ATG5, ATG3, ATG7, ATG10, ATG12, beclin1, γ-aminobutyric acid receptor related protein (GABARAP), p62, and NBR1 [7,10]. In addition, the activated IRE1 promotes the splicing of XBP1 mRNA, and the spliced XBP1 triggers the autophagy signal pathway by regulating the transcriptional of BECN1 [71].

### 4.4. Chaperone-Mediated Autophagy

Chaperone-mediated autophagy (CMA) is another type of autophagy that degrades cytoplasmic proteins in lysosomes without autophagy-related gene. Hsc70 forms complexes with Hsp40, Hip, Hop, and Hsp90 to recognize and bind substrates with KFERQ-like sequences [72]. Then the unfolded substrates interact with molecular chaperone complex and then are transferred to the lysosomal cavity by binding Lysosome-Associated Membrane protein type-2A (LAMP-2A). Thus, LAMP-2A is a rate-limiting factor for CMA. After entering the lysosome, the substrates are rapidly degraded by lysosomal proteases, and then chaperone complex is released from the lysosomal membrane. Starvation and serum withdrawal increase the LAMP-2A protein level and its reinsertion into the lysosomal membrane by reducing its degradation. After long-term starvation, 30% of the cytoplasmic proteins are degraded through the CMA pathway. LAMP-2A identifies membrane-associated ring-CH-type finger 5 (MARCHF5) through KFERQ-like consensus, an E3 ubiquitin ligase required for mitochondria fission. Severe oxidative stress compromised CMA activity and stabilized MARCHF5, which facilitated Dynamin-1-Like (DNM1L) translocation and led to excessive mitochondria fission [72]. Furthermore, a report shows that CMA is required for PQC in stem cells and for the upregulation of fatty acid metabolism upon hematopoietic stem cells (HSCs) activation [73].

## 5. Crosstalk between UPS and Autophagy

### 5.1. UPR Mediates Crosstalk between UPS and Autophagy

UPS and autophagy are two major ways of protein degradation in cells. More and more studies have shown a coordinated and complementary relationship between these two systems [74,75]. When misfolded proteins cannot be degraded by the molecular chaperone-mediated proteasome pathway and cause protein aggregation, which may in turn inactivate proteasome and lead to cytotoxicity, the autophagy lysosome pathway is an important compensation mechanism for mediating the degradation of ubiquitinated protein aggregates [63].

UPR plays an important role in the process of autophagy induced by proteasome inhibitor. Due to the destruction of proteasome, the misfolded proteins cannot be degraded by proteasome and form aggregates, which destroys the homeostasis of endoplasmic reticulum and induces UPR [76]. For example, proteasome inhibitor activates autophagy through the IRE/JNK/bcl2/BECN1 axis [77,78] (Figure 1). Activated IRE1 can also recruit TRAF2, resulting in JNK phosphorylation and the expression of autophagy core gene [79,80]. At the same time, ATF6 is cleaved and transferred from the Golgi apparatus to the nucleus, where it induces the expression of death-associated protein kinase 1 (DAPK1), [81] and ATF6 can also promote the biogenesis of autophagosomes by increasing phosphorylation of BECN1. Concurrently, many studies have shown that proteasome inhibitors can increase the expression of ATF4 through PERK axis, and previous studies proved that ATF4 regulates the transcription of autophagy-related genes, such as LC3, ATG5, and ATG7. Proteasome inhibitors can also inhibit the ubiquitin degradation of ATF4 [82].

### 5.2. Proteasome Inhibitors Regulates Proteins Involved in Autophagy

In addition to UPR, proteasome inhibitors can also directly act on autophagy-related proteins to promote autophagy [63]. For example, proteasome inhibitor destroys the protein–protein interaction between Raptor, one of the structural components of mTORC1, and its interacting partners, thereby inhibiting the activity of mTORC1 [83,84] (Figure 1). In addition, p53 accumulates and is transferred to the nucleus and acts as a transcription factor for autophagy housekeeping genes after inhibiting the proteasome, such as diastasis of the rectus abdominis muscle (DRAM) [81]. Alternatively, elevation of p53 level may activate autophagy by inhibiting the mTORC1 pathway [83] (Figure 1). Furthermore, a number of studies have shown that the protein level of LC3 is significantly increased when tumor cells are exposed to proteasome inhibitors [85,86].

Moreover, p62 is an important bridge between the proteasome system and autophagy [87]. There is evidence that proves that proteasome inhibitors and starvation can promote the synthesis of p62 and impair the protein toxicity stress caused by protein aggregation [88]. When the proteasome system is inhibited, E3 ligase TRIM50 co-locates with p62 and HDAC6 on aggregates or ubiquitin aggregate proteins to help autophagy machines to recognize aggregates and remove them [89]. The transcription of p62 is mainly regulated by NRF2. In the basic state, NRF2 is degraded by the Cul3-Keap1-E3ligase (keap1)-mediated ubiquitin–proteasome pathway. Under the stimulation of oxidative stress, the ubiquitination of NRF2 by Keap1 is blocked; thus, NRF2 is transferred to the nucleus to promote the transcription of p62, while overexpressed p62 competes with NRF2 to bind KEAP1, thus forming a positive feedback regulatory loop [90]. Under cellular stress, ULK1 can phosphorylate the serine 405 and 409 of p62, thus increasing the affinity of p62 to ubiquitin proteins and leading to the effective degradation of protein aggregates. The inhibition of proteasome can also activate the adaptive transcription of NRF2, which, in turn, promotes the synthesis of p62 [91].

HDAC6 is another key protein in linking UPS and autophagy. HDAC6 regulates the acetylation of α-tublin, which not only participates in the formation of aggregates but also promotes the transport of ubiquitin aggregates to new phagocytic vesicles [9,92,93]. HDAC6 interacts with polyubiquitinated proteins and brings these proteins to the dynamins, which transport polyubiquitinated proteins to the autophagosomes [9,94]. HDAC6 is involved in the fusion of autophagosomes and lysosomes, and inhibition of HDAC6 can enhance the cytotoxicity of proteasome inhibitors [93].

In addition, Bcl-2-associatedathanogene (BAG) family proteins also mediate the crosstalk of the proteasome pathway and autophagy. BAG1 promotes proteasome degradation of unfolded proteins by interacting with Hsc70/Hsp70 chaperone complexes and proteasomes. Meanwhile, BAG3 interacts with the complex composed of heat-shock protein B8 (HSPB8) and Hsp70, which recruit p62, and substrates bind with Hsp70 to LC3 [95]. In senescent cells or tissues, the conversion of BAG1 expression to BAG3 during inhibition of proteasome or oxidative stress leads to autophagy as an alternative pathway for PQC [96,97].

### 5.3. Inhibition of Autophagy Affects UPS

Inhibition of proteasome leads to a compensatory stimulation of autophagy, whereas the inhibition of autophagy activates or impairs proteasomal flux via several mechanisms. For example, overabundant p62 delays the transport of substrates to proteasomes [98]. Moreover, research shows that USP10 interacts with p62 and augments p62-dependent ubiquitinated protein aggregation and aggresome formation, which inhibit proteasome activity [99]. Another possible mechanism of autophagy inhibition impacts proteasome system is proteaphagy. Senescent or inactivated proteasomes and subunits are eliminated by selective autophagy to maintain the balance of the overall proteasome pool, which is known as proteaphagy. When proteaphagy is inhibited, senescent or inactivated proteasomes increase and compete with normal proteasome, and this affects the proteasome flux [10]. However, there is an opposite view that proteasome can be activated when autophagy is inhibited by drugs or gene editing. After knocking out autophagy-related protein factors PIK3C3, ATG5, and ATG7, the activities of three types of proteasome are upregulated, and the protein levels of proteasome subunits, including proteasome β 5 subunit (PSMB5), are also increased.

## 6. UPS and Autophagy in Disease

### 6.1. Neurodegenerative Disease

In recent years, many clinical studies have shown that UPS and autophagy are closely related to many diseases. The dysfunction in regard to the elimination of misfolded or aggregated proteins from the cytoplasm of neurons and glial cells seems to be a particularly common cause of various neurodegenerative diseases [6,100]. In the past decade, a number of studies have shown that ER plays a key role in the pathogenesis of neurodegenerative diseases, including Alzheimer’s disease (AD), amyotrophic lateral sclerosis (ALS), Parkinson’s disease (PD), and hypermetabolic disease (GD). Parkinson’s disease is a progressive neurodegenerative disease characterized by α-synuclein (α-syn) oligomers, but not monomers or fibers, suggesting the unique ability of α-syn oligomers to perturb cellular processes [101,102]. Oligomeric α-syn uniquely leads to ER stress. The two main ways to maintain proteostasis are UPS and autophagy, which are both affected by α-syn oligomers. Oligomers inhibit proteasome activity and cause lysosome dysfunction [102]. Moreover, α-Syn is highly susceptible to ubiquitination, and dysfunction in α-Syn ubiquitination contributes to chronic synucleinopathies [103].

In the past few decades, a large number of studies have improved our understanding of the complexity of ubiquitin networks. In view of the abnormality of E3 ligase in a variety of pathogenic processes, E3 ligases have become an effective target for the treatment of diseases. For example, Parkin is an ubiquitin E3 ligase involved in a variety of cellular processes associated with PD [104]. One of Parkin’s targets for ubiquitination, Parkin interacting substrate (PARIS) accumulates in the brain of patients with autosomal recessive juvenile PD [105]. Mutations to Parkin result in the accumulation of PARIS and successive inhibition of the transcription of Peroxisome proliferator-activated receptor gamma coactivator 1-alpha (PGC-1α) and its downstream targets, thus affecting the degradation of damaged mitochondria and biogenesis of mitochondrial. Autosomal recessive juvenile parkinsonism (AR-JP) is related to the mutation of Parkin [106]. In addition, the overexpression of Parkin specifically inhibits UPR-induced cell death, but the mutant Parkin cannot.

There is much evidence for the association of dysregulation of ubiquitin-mediated autophagy with various neurodegenerative diseases. For example, the wild ataxin 3 in spinocerebellar ataxia type 3 is a DUB of BECN1. With this function, ataxin 3 is required for starvation-induced autophagy [107]. Ataxin 3 with expanded polyQ repeats has higher binding affinity with BECN1, which is defective in removing ubiquitin chain from BECN1. These findings identify a link of ataxin 3 to autophagy regulation, and, more important, impairment of Beclin-1 mediated autophagy accounts for one mechanism of polyQ repeat-associated neurodegenerative diseases. USP19-depedent Beclin-1 deubiquitination mediates selective autophagy and anti-inflammation and determines the outcome of infection and immunological functions [108]. TAX1BP1, a particular autophagy receptor, also possesses a unique role in cell intrinsic control of infection [109].

### 6.2. Cancers

#### 6.2.1. UPR and Cancers

In addition to neurodegenerative diseases, PQC is closely related to tumor development, invasion, and drug resistance [110]. In the process of carcinogenesis, tumor cells are exposed to various stresses, such as lack of nutrition, accumulation of acid waste and hypoxia, change of chromosome number, activation of oncogenes, inactivation of tumor suppressor genes, and accelerated secretion, leading to endoplasmic reticulum stress and then activating the PQC mechanism of endoplasmic reticulum. First of all, three UPR signaling pathways will block tumor development in the early stage of cancer, while cancer cells will adapt to internal and external stress and resist apoptosis caused by endoplasmic reticulum stress in later stages [111]. For example, PERK is highly expressed in a variety of tumor cells, such as kidney renal papillary cell carcinoma, brain lower-grade glioma, breast-invasive carcinoma, and thyroid carcinoma, and the high expression of PERK is associated with poor prognosis, while the high expression in neck squamous cell carcinoma is well correlated with good prognosis [112]. PERK-mediated upregulation of vascular endothelial growth factor (VEGF), fibroblast growth factor-2 (FGF2), and interleukin-6 (IL-6) and downregulation of anti-angiogenic cytokines significantly promoted tumor growth [113,114]. However, Ajda Coker-Gurkan et al. demonstrated that Atiprimod upregulated the expression of Bak, Bax, and Bim through the PERK/eIF2 α/ATF4/CHOP pathway and promoted the apoptosis of breast cancer cells [115]. XBP1 promotes the occurrence, development, and recurrence of triple-negative breast cancer by regulating the transcription of HIF1-α [116]. In addition, constitutive activation of XBP1 in tumor-associated DCs (tDCs) drives ovarian cancer (OvCa) progression by blunting antitumor immunity [117]. Recent studies have shown that XBP1s protein can upregulate the expression of insulin-like growth factor binding protein-3 (IGFBP3) and regulate the invasion and metastasis of NSCLC cells by regulating IGFBP3 [118]. Speaking of ATF6, Jiao Meng et al. have shown that activated STAT3 promotes the transcription of ATF6, which, in turn, endows cancer cells with resistance to cisplatin and paclitaxel [119]. DAPK1 is a metastasis inhibitory factor, whose mechanism is to inhibit tumor metastasis and mediate apoptosis and autophagy. Downregulation of DAPK1 expression was detected in ATF6 knockout cells, which affected the expression of ATG9 to regulate autophagy flux. At the same time, ATF6-mediated upregulation of CHOP also contributes to ATF6-induced autophagy [81]. In colon cancer, XBP1 and ATF6 activation reduced cellular proliferation and reduced expression of markers of intestinal epithelial stemness [120].

#### 6.2.2. UPS and Cancers

Alterations in the UPS have the potential to alter cellular homeostasis, and studies have demonstrated that proteasomal activity is elevated in human cancers. This may be due to the fact that UPS regulates key proteins in many cellular processes, such as epithelial–mesenchymal transition (epithelial–mesenchymal transition, EMT), cell cycle, signal transduction, gene expression, DNA repair, and apoptosis. A study has shown that, in cyclosporine A (CsA)-treated cells, there is an increased expression of PPIL2, a ubiquitin E3 enzyme, which ultimately leads to the decrease of SNAI and thus inhibits the EMT process of breast cancer cells [121]. UPS also degrades key proteins in the signaling pathway in response to cellular stress, such as, Wnt/β-catenin, HIF-α, and p53. Here we summarize the signal pathways related to tumorigenesis regulated by the ubiquitin–proteasome pathway in Table 2.

#### 6.2.3. Autophagy and Cancer

Inducing autophagy in cancer may be helpful in cancer treatment [55]. In fact, autophagy has a dual role in cancer that is related to the type, stage, or genetic context of the cancers [143,144,145]. In the study of melanoma, small-molecule HA15 induces apoptosis and autophagy in vivo and in vitro by targeting the activation of GRP78, a landmark protein of endoplasmic reticulum stress [146]. The occurrence of autophagy is accompanied by the accumulation of vesicles, the transformation of LC3-I to LC3-II, and the formation of autophagosomes [146]. The therapeutic effect of HCA15 on melanoma cells decreases with the decrease of autophagy and apoptosis, suggesting that autophagy can inhibit tumor growth. Moreover, inhibition of autophagy can also inhibit the invasion and migration of tumor cells [147,148,149]. However, some studies have shown that autophagy can mediate the immune and chemotherapy resistance of cancer [150]. Cancer stem cells (CSCs) in a quiescent state obtain resistance to conventional treatment; this is the main cause of tumor recurrence. Chemotherapy drugs promote autophagy in tumor cells. However, the combination of chemotherapy drugs and autophagy inhibitors can make CSCs sensitive to chemotherapy drugs. CSCs heterogeneity and patient-specificity make the situation more complicated. We are far from finding new drug combinations that would allow us to eradicate CSCSs or at least inhibit their proliferation. In addition, several studies have shown that some HER2-positive breast cancer cells can acquire drug resistance through the upregulation of autophagy pathways [151,152]. Due to the dual role of autophagy in tumors, whether autophagy can be used as a therapeutic target still needs further study.

## 7. Conclusions

Under external stimuli, protein denature and aggregate are harmful to the cells. Ubiquitin can protect cells from the stimuli by playing a key role in the clearance of protein aggregates. Therefore, maintaining the homeostasis of ubiquitin pools is very crucial for protein quality control, and transcription factors which respond to external stimuli and deubiquitination enzymes play an important role in ubiquitin pool homeostasis. In the presence of ubiquitin monomer, the proteins that need to be degraded in the cells will be marked by ubiquitin. Ubiquitin-labeled protein aggregates are ultimately degraded by UPS or autophagy. The ubiquitin, UPS, and autophagy are the interdependent elements of the protein quality-control system; they must act in a networked manner to maintain proteostasis. Meanwhile, many signal pathways are involved in this. A better understanding of individual systems, as well as the interconnections and crosstalk between them, is beneficial for clinical management of diseases involving PQC problems. Meanwhile, UPS and autophagy have been proved to have different functions in different diseases. For example, autophagy inhibits or promotes cancer depending on the type of cancer [143,144,145,146,147,148,149,150,151,152], thus making it is difficult and complicated to find the precise therapeutic targets for different diseases with different PQC problems.

## Figures and Tables

**Figure 1 cells-11-00851-f001:**
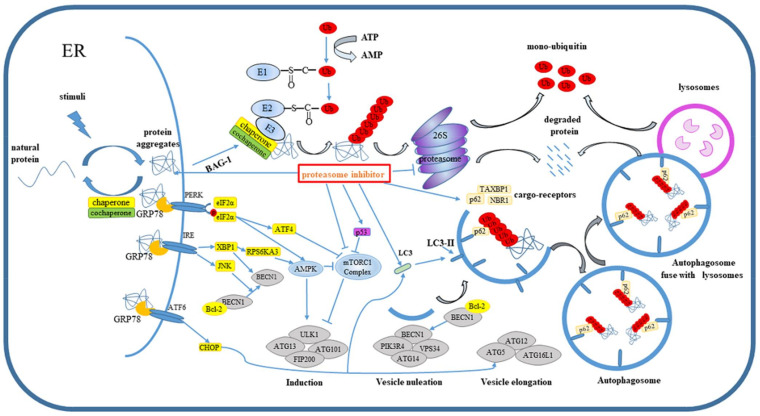
Ubiquitin mediates the degradation of protein aggregates under stimuli. In response to cell stimulation, proteins in the endoplasmic reticulum form aggregates, which are labeled by ubiquitin in the presence of E1, E2, and E3 and then degraded by proteasome or autophagosome. UPR mediated by PERK, IRE, and ATF6 possesses a vital role in regulating autophagy and UPS. Unc-51-like kinase 1 (ULK1), Autophagy-Related 13 (ATG13), RB1-inducible coiled-coil protein 1 (FIP200), neighbor of *BRCA1* gene 1 (NBR1), Beclin-1 (BECN1), 5′-AMP-activated protein kinase (AMPK), mTOR complex 1 (mTORC1), ribosomal protein S6 kinase (RPS6KA3), Tax-binding protein 1 (TAXBP1), phosphoinositide 3-kinase (PI3K), phosphoinositide-3-Kinase Regulatory Subunit 4 (PIK3R4), and Phosphatidylinositol 3-Kinase Catalytic Subunit Type 3 (vps34). UPS: glycine at the C-terminus of ubiquitin binds to the active site of E1, along with the hydrolysis of ATP. the active ubiquitin is then transferred to the E2 enzyme to form a complex with the E2 conjugating activation, and this complex further interacts with the specific enzyme E3, which makes the ubiquitin transfer to the specific substrate. Autophagy lysosomal system (ALS) induction: in response to cell stress, (such as starvation), the complex formed by ULK1, mAtg13, Atg101, and FIP200 is activated to initiate the de novo isolation of a portion of intracellular membrane. Vesicle nucleation: ULK1 complex phosphorylates autophagy protein BECN1 and promotes it to form a class PI3K complex (including PIK3R4, BECN1, ATG14, and VPS34). Vesicle elongation: ATG16L1 binds with ATG5–ATG12 to form the active ligase-like complex ATG12–ATG5–ATG16L1, serve as E3 ligase, and make LC3-I combined with PE to form LC3-PE (LC3-II). LC3-II anchors to the autophagy vesicle and promotes the extension of autophagy vesicle. Autophagosome: Cargo receptors of autophagy (such as p62, NBR1, and TAXBP1) transport the ubiquitinated substrate to the autophagosome. Finally, autophagosomes fuse with lysosomes. UPR regulates UPS and ALS to clear protein aggregates in response to cellular stress.

**Table 1 cells-11-00851-t001:** Ubiquitin E3 ligase in protein quality control under stimuli.

Ubiquitin E3 Ligase	Substrates
GP78	CD3—δ [22], CFTR [23], HMGCR [24].
Parkin	α-synuclien, synphilin-1, Pae1 [25], glucocerebrosidase (GCase) [26].
E3 ubiquitin–protein ligase synoviolin (HRD1/3) [21,27]	Hmg1p [28], pre-B cell receptor (pre-BCR), pro-arginine vasopressin (AVP) B-lymphocyte-induced maturation protein 1 (BLIMP1) [29], NRF2 [30], peroxisome proliferator activated receptor γ coactivator-1 β (PGC1β) [31], transforming growth factor beta (TGF-β) [32], T-cell receptor alpha (TCR-α), [33] alpha1-antitrypsin [34], p53 [35], amyloid precursor protein (APP) [36], et al.
membrane-associated RING C3HC4 finger 6 (MARCHF6) [37,38,39]	squalene epoxidase (SQLE) [40], HMGCR [38], perilipin-2 (PLIN2) [38], et al.
RNF5 [41]	CFTR [41], JNK-associated membrane protein (JAMP) [42], et al.
Makorin ring-finger protein 1 (MKRN1)	Potassium voltage-gated channel subfamily H member 1 (KCNH1), Eag1 [43]
SCF ^Fbx2^ [37]	β-secretase (BACE1) [44], CFTR [45], et al.
SCF ^Fbx6^ [37]	Chk [46]
SCF^β-TrCP1/2^	CD4 [47], Tetherin [48]
CHIP [24]	(CFTR) [19,20,21], GR [22], Pael [23], ErbB2 [23], IRE [49], INO80 [50], tau [51], et al.
SMAD ubiquitination regulatory factor 1 (Smurf1)	Wolframin (WFS1) [52], phosphatase and tensin homolog (PTEN) [53], et al.
Neuregulin receptor degradation pathway protein 1 (Nrdp1) [37]	Erb-B2 receptor tyrosine kinase 3 (ErbB3) [54], et al.

**Table 2 cells-11-00851-t002:** Signaling pathways regulated by the ubiquitin–proteasome pathway under cell stimuli.

Signal Pathway	Cell Stimuli	E3 Enzyme
NF-κB pathway [111,112]	Cytokines, UV, virus, oxide, et al.	β-TrCP [122], IRF3 [123], Linear Ubiquitin Chain Assembly Complex (LUBAC) [124]
p53 pathway	DNA damage	MDM2 [125,126], FBXW7 [127]
DDB1/DDB2-XPC/HR23B pathway	DNA damage	DDB1-CRBN ubiquitin E3 ligase [128]
MAPK pathway [129]	UV, Osmotic pressure change, heat shock, et al.	TRIM48 [130], NEDD4 [123], RNF [131,132], et al.
HIFα pathway [133]	hypoxia	pVHL E3 ligase complex [134].
wnt/β-catein pathway	DNA damage, hypoxia, virus, et al.	SIAH1 [135,136,137].
PI3K/AKT/mTOR pathway	Hypoxia, high glucose, cytokines, et al.	FBXL12 [138], parkin [139,140], TRAF6 [141,142], β-TRCP [143], et al.

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
