# Peer review of "Ubiquitination-Proteasome System (UPS) and Autophagy Two Main Protein Degradation Machineries in Response to Cell Stress"

_cells, 2022, doi:10.3390/cells11050851_

Round 1

Reviewer 1 Report

In this manuscript, the authors nicely summarized the current knowledge and our understanding of crosstalk between ubiquitin proteasome system and autophagy. It is written very thoroughly. I have only a few minor suggestions to further improve this manuscript. 

  1. The title of the paper is not clear. I would suggest: Ubiquitin-proteasome system (UPS) and autophagy: two main protein degradation machineries in response to cell stress. /or: The crosstalk between UPS and autophagy in response to cell stress/or: The role of ubiquitination in mediating autophagy in in response to cell stress.
  2. Introduction section, the authors should highlight that the two systems are operated independently, however recent studies have revealed multiple layers of interconnections between UPS and autophagy. For instance, inhibition of UPS leads to a compensatory stimulation of autophagy via several mechanisms, whereas autophagy inhibition activates or impairs proteasomal flux depending on the cellular and environmental conditions [1, 2]. In addition, components of either system can serve as the proteolytic targets of the other system [2].

Ref: 1. Ji CH, Kwon YT. Crosstalk and interplay between the ubiquitin-proteasome system and autophagy. Mol Cell. 2017;40:441–9.

2, Kocaturk NM, Gozuacik D. Crosstalk between mammalian autophagy and the ubiquitin-proteasome system. Front Cell Dev Biol. 2018;6:128.

   3. In section 6: Ubiquitin and disease, it would be better to address more details about how dysregulation of ubiquitin-mediated autophagy pathways is associated with human diseases not limited to cancers

Author Response

Reviewer 1

In this manuscript, the authors nicely summarized the current knowledge and our understanding of crosstalk between ubiquitin proteasome system and autophagy. It is written very thoroughly. I have only a few minor suggestions to further improve this manuscript.  

Point 1: The title of the paper is not clear. I would suggest: Ubiquitin-proteasome system (UPS) and autophagy: two main protein degradation machineries in response to cell stress. /or: The crosstalk between UPS and autophagy in response to cell stress/or: The role of ubiquitination in mediating autophagy in in response to cell stress.

Response 1: I changed the original title: Ubiquitination: a key determinator of protein fate through proteasome degradation or autophagy in response to cell stress into Ubiquitin-proteasome system (UPS) and autophagy: two main protein degradation machineries in response to cell stress.

Point 2: Introduction section, the authors should highlight that the two systems are operated independently, however recent studies have revealed multiple layers of interconnections between UPS and autophagy. For instance, inhibition of UPS leads to a compensatory stimulation of autophagy via several mechanisms, whereas autophagy inhibition activates or impairs proteasomal flux depending on the cellular and environmental conditions [1, 2]. In addition, components of either system can serve as the proteolytic targets of the other system [2]. Ref: 1. Ji CH, Kwon YT. Crosstalk and interplay between the ubiquitin-proteasome system and autophagy. Mol Cell. 2017;40:441–9.  2, Kocaturk NM, Gozuacik D. Crosstalk between mammalian autophagy and the ubiquitin-proteasome system. Front Cell Dev Biol. 2018;6:128.

Response 2: I added the following sentences to the introduction Ubiquitin is a classical molecular label for protein degradation. When the misfolded proteins and unfold proteins are tagged with ubiquitin, these proteins will be degraded in proteasome or lysosome [8, 9]. Ubiquitin not only plays a role in UPR, but also participates in the activation of cellular stress responses. UPS and autophagy are two indenpendly system in protein quality control (PQC).However, there is more evidence about the con-nections between UPS and Autophagy. In this review, we first briefly summarize the ubiquition pool ,UPS and Autophagy, and then discuss in detail various examples of co-ordination and crosstalk between them and dysregulation of UPS and ubiquitin-mediated autophagy pathways in human diseases, neurodegenerative diseases and cancer in particular.

Point 3. In section 6: Ubiquitin and disease, it would be better to address more details about how dysregulation of ubiquitin-mediated autophagy pathways is associated with human diseases not limited to cancers

Response 3: I added the details of dysregulation of ubiquitin-mediated autophagy in neurodegenerative diseases in 6.1 neurodegenerative disease.

“There’s many evidence for the association of dysregulation of ubiquitin-mediated autophagy with various neurodegenerative disease. For example, the wild ataxin 3 in spinocerebellar ataxia type 3 is a DUB of BECN1.With this function, ataxin 3 is required for starvation-induced autophagy[112]. Ataxin 3 with expanded polyQ repeats has higher binding affinity with BECN1, which is defective in removing ubiquitin chain from BECN1. These findings identify a link of ataxin 3 to autophagy regulation and, more importantly, impairment of Beclin-1 mediated autophagy accounts for one mechanism of polyQ re-peat-associated neurodegenerative diseases. USP19-depedent Beclin-1 deubiquitination mediated selective autophagy and anti-inflammation and determine the outcome of infection and immunological functions[113]. TAX1BP1 , a particular autophagy receptor , also posseses a unique role in cell intrinsic control of infection[114].”

Reviewer 2 Report

In this manuscript, Y Li and colleagues sough to make a literature review of the role of ubiquitination in regulating protein degradation via the proteasome or autophagy in response to stresses.

The review lacks originality and present numerous flaws, starting with reducing ubiquitination to a way to target proteins for degradation. The topic of the proposed review is very broad and appears to lack some focus as it only skims through matters related to the UPS, UPR, autophagy, relation to diseases.

Further specific comments:

  1. The figure 1 is very complex and would benefit from being clarified. Alternatively, it could be split in sub-figures.
  2. The table 2 seems very restrictive/selective, as virtually every cellular process is regulated by ubiquitination and degradative pathways (UPP and autophagy)
  3. The manuscript should be proof-read as there are numerous grammatical, spelling and syntaxis mistakes along the text (only a few listed below).
  4. Paragraph 4.3: this paragraph is fairly tedious to read due to the listing of proteins. Would a figure help to clarify it?
  5. Line 72: about ubiquitin chains, the authors should also mention the other chain types.
  6. Line 78: DUBs do not only have a function in degradation processes
  7. Line 433: it’s rather odd to finish a conclusion on “which is very complicated and difficult” without proper explanation.
  8. Line 336: what do the authors mean by “imbalance of E3”?
  9. Line 65: “the quality of protein in and” -> “in”?
  10. Line 67: “protein molecule” is redundant
  11. Line 68: founded -> found
  12. Lines 101/102: what do the authors mean by “unnatured” or “unnatural”? These terms come back several time along the manuscript.
  13. Lines 119-120: chaperone -> chaperoneS, segment -> segmentS
  14. Lines 120-121: “and lead to misfolded proteins to refold…” the sentence is not clear
  15. Line 26: substrateS
  16. Line 137: “ubiquitination degradation” should read as “ubiquitin-mediated degradation”?
  17. Line 151: aggrephagy is only one example of many of selective autophagy
  18. Line 171: proteaseS, sent
  19. Line 183: “autophagosomeand” space missing
  20. Line 294: “Bcl-2-associatedathanogene” space missing
  21. Lines 304-306: first sentence of the paragraph is unclear
  22. Lines 306, 307: interactS, augmentS
  23. Line 312: “the increase of aging or…” this sentence is unclear
  24. Line 335: improveD
  25. Line 337: E3 ligaseS HAVE
  26. Lines 356, 395: “et al”?
  27. Lines 378, 410: repetition of the protein name
  28. Line 412: “Chemotherapy drugs are known…” this sentence is unclear
  29. Line 422: what do the authors mean by “protein denature”?

Author Response

Point 1:In this manuscript, Y Li and colleagues sough to make a literature review of the role of ubiquitination in regulating protein degradation via the proteasome or autophagy in response to stresses.  The review lacks originality and present numerous flaws, starting with reducing ubiquitination to a way to target proteins for degradation. The topic of the proposed review is very broad and appears to lack some focus as it only skims through matters related to the UPS, UPR, autophagy, relation to diseases. 

Response1: In this review, we first briefly summarize the ubiquition pool ,UPS and Autophagy, and then discuss in detail various examples of coordination and crosstalk between them and dysregulation of UPS and ubiquitin-mediated autophagy pathways in human diseases, neurodegenerative diseases and cancer in particular.

Further specific comments: 

Point 2:The figure 1 is very complex and would benefit from being clarified. Alternatively, it could be split in sub-figures.

Response2: I wanted to show the relationship between cell stress, ubiquitin, UPS and autophagy, and the contents shown in Figure 1 are distributed throughout the whole article. Therefore, it is better not to split Figure 1.I added some explanation to Figure 1. “UPS: glycine at the C-terminus of ubiquitin binds to the active site of E1, along with the hydrolysis of ATP. the active ubiquitin is then transferred to the E2 enzyme to form a complex with the E2 conjugating activation, and this complex further interacts with the specific enzyme E3, which makes the ubiquitin transfer to the specific substrate. Autophagy lysosomal system(ALS): Induction: in response to cell stresss (such as starvation), the complex formed by ULK1, mAtg13, Atg101 and FIP200 is activated to initiate the de novo isolation of a portion of intracellular membrane; Vesicle nucleation: ULK1 complex phosphorylates autophagy protein BECN1 and promotes it to form a class PI3K complex (including PIK3R4, BECN1, ATG14, VPS34);Vesicle elongation: ATG16L1 binds with ATG5-ATG12 to form the active ligase-like complex ATG12-ATG5-ATG16L1, serves as E3 ligase,and make LC3-I combined with PE to form LC3-PE (LC3-II). LC3- II anchors to the autophagy vesicle and promotes the extension of autophagy vesicle; Autophagosome: Cargo receptors of autophagy(such as p62,NBR1,TAXBP1) transport the ubiquitinated substrate to the autophagosome. Finally, autophagosomes fuse with lysosomes. UPR regulates UPS and ALS to clear protein aggregates in response to cellular stress.”

Point 3: The table 2 seems very restrictive/selective, as virtually every cellular process is regulated by ubiquitination and degradative pathways (UPP and autophagy) The manuscript should be proof-read as there are numerous grammatical, spelling and syntaxis mistakes along the text (only a few listed below).

Response 3: In fact, there are many ubiquitin E3 enzymes involved in protein degradation. Due to space constraints, we only summarize the ubiquitin E3 enzymes responsible for the degradation of misfolded proteins and protein aggregates.

Point 4: Paragraph 4.3: this paragraph is fairly tedious to read due to the listing of proteins. Would a figure help to clarify it?

Response 4: The relationships of these proteins are summarized in figure1. Maybe I'm not being specific or clear enough. I redescribed it.

Point 5: Line 72: about ubiquitin chains, the authors should also mention the other chain types.

Response 5. ubiquitin moieties can be conjugated through one of their lysine residues (K6, K11, K27, K29, K33, K48 and K63) or the N-terminal methionine residue (M1).In addition to polyubiquitination, there are monobituitination and multi-monobituitination . Polyubiquitination can also be divided into homotypic polyubiquitination and heterotypic   polyubiquitunqition [1].

Point 6: Line 78: DUBs do not only have a function in degradation processes

Response 6: DUBs, meanwhile, maintains protein stability, edits ubiquitin chains and processes ubiq-uitin precursors and removes non degradative ubiquitin signal.

Point 7: it’s rather odd to finish a conclusion on “which is very complicated and difficult” without proper explanation.

Response 7: Because the ubiquitin, UPS and autophagy are the interdependent elements of the protein quality control system, they must act in a networked manner to maintain proteostasis. Meanwhile, many signal pathways involved in this. A better understanding of individual systems as well as the interconnections and crosstalks between them is benifical for clinical management of diseases involving protein quality control problems. While UPS and autophagy has been proved has different function in defferent diseases. For example, autophagy  inhibits or promotes cancer depends on the types of can-cer[147-152], which makes it is difficult and complicated to find the precise therapeutic targets for different diseases with different PQC problems.

Point 8Line 336: what do the authors mean by “imbalance of E3”?

Response 8. I change “imbalance into abnormality”. The abnormality of E3 include mutation of E3 as I illustrate below.

Point 9: Line 65: “the quality of protein in and” -> “in”?

Response 9: I delete “in”.

Point 10: Line 67: “protein molecule” is redundant

Response 10: I changed “Ubiquitin is a highly conserved small protein molecule containing 76 amino acids…….” into “Ubiquitin is a highly conserved,containing 76 amino acids, ……”

Point 11: Line 68: founded -> found

Response 11: Notice the subject. Here, we should use the Passive Voice instead of Active Voice. I don’t change the word.

Point 12: Lines 101/102: what do the authors mean by “unnatured” or “unnatural”? These terms come back several time along the manuscript.

Response 12: I changed “unnatured” into “unnatural” and unnatural protein means misfolded and unfolded protein. I've seen this description in references.

Point 13: Lines 119-120: chaperone -> chaperoneS, segment -> segmentS

Response 13: I changed “chaperone” into “chaperones” and I deleted segment.

Point 14: Lines 120-121: “and lead to misfolded proteins to refold…” the sentence is not clear

Response 14: I change the sentence “Molecular chaperone and peptide segment exposed hydrophobic patches to promote new peptides folding, and lead to misfolded proteins to refold to inhibit the aggregation of protein under cellular stress.”into “Chaperones and proteases play a key role of protein quality control (PQC) when protein aggregates occurs. Molecular chaperones promote the folding of new peptides and refold-ing of misfolded proteins to inhibit the aggregation of protein through recognition of hy-drophobic patches of misfolded protein and unfolded proteinunder cellular stress.”

Point 15: Line 26: substrateS

Response 15: I’m sorry I did not find the words “substrate”.

Point 16: Line 137: “ubiquitination degradation” should read as “ubiquitin-mediated degradation”?

Response 16: I changed “ubiquiytination degradation” into “ubiquitin-mediated degradation”.

Point 17: Line 151: aggrephagy is only one example of many of selective autophagy

Response 17: I agree with the reviewer. What I want to show here is that protein aggregates can be removed by selective autophagy. I changed the sentence to make it easier to understand.

Point 18: Line 171: proteaseS, sent

Response 18: I changed “protease” into “proteases”, “send” into“sent”.

Point 19: Line 183: “autophagosomeand” space missing

Response19: I changed “autophagosomeand” into “autophagosome and”.

Point 20: Line 294: “Bcl-2-associatedathanogene” space missing

Response 20: I changed “Bcl-2-associatedathanogene” into “Bcl-2-Associated Athanogene”.

Point 21: Lines 304-306: first sentence of the paragraph is unclear.

Response 21: I changed the first sentence into “Inhibition of proteasome leads to a compensatory stimulation of autophagy via several mechanisms, whereas autophagy inhibition activates or impairs proteasomal flux. For example, overabundant p62 delays the transport of proteasome substrate to proteasome”.

Point 22:Lines 306, 307: interactS, augmentS

Response 22: I changed “interact” into “interacts”, “augment” into “augments”.

Point 23: Line 312: “the increase of aging or…” this sentence is unclear

Response 23: I changed the original sentence “When proteaphagy is inhibited, the increase of aging or inactivated proteasome competes with normal proteasome, which affect proteasome flux.” into “When proteaphagy is inhibited, senescent or inactivated proteasomes increase and com-pete with normal proteasome, which affect proteasome flux.”

Point 24 :Line 335: improved

Response 24: I changed “improve” into “improved”.

Point 25: Line 337: E3 ligaseS HAVE

Response 25: I changed “E3 ligase has” into “E3 ligases have”

Point 26: Lines 356, 395: “et al”?

Response 26: I delete “et al”

Point 27: Lines 378, 410: repetition of the protein name

Response 27: I changed “Death-associated protein kinase1 (Death-associated protein kinase1, DAPK1)”, “(Cancer stem cells, CSCs)” into “(CSCs)”.

Point 28: Line 412: “Chemotherapy drugs are known…” this sentence is unclear

Response 28: I change the original sentence “Chemotherapy drugs are known to cause tumor cells to autophagy” into “Chemotherapy drugs promote autophagy in tumor cells”.

Point 29: Line 422: what do the authors mean by “protein denature”?

Response 29: protein denaturation: Due to the influence of physical or chemical factors, the original specific structure of natural protein molecules is changed, which results in partial or complete loss of its properties and functions. 

Round 2

Reviewer 2 Report

In their revised version of the manuscript and the associated letter, Li and colleagues have considered all my previous comments.  The quality of the manuscript has been significantly improved.

However, it would be essential that the manuscript undergoes extensive proofreading by a native English speaker or subscribe to a proofreading service as there are many mistakes in the grammar and syntax.